# Modelling Population Genetic Screening in Rare Neurodegenerative Diseases

**DOI:** 10.3390/biomedicines13051018

**Published:** 2025-04-23

**Authors:** Thomas P. Spargo, Alfredo Iacoangeli, Mina Ryten, Francesca Forzano, Neil Pearce, Ammar Al-Chalabi

**Affiliations:** 1Department of Basic and Clinical Neuroscience, Maurice Wohl Clinical Neuroscience Institute, King’s College London, London WC2R 2LS, UK; 2Department of Biostatistics and Health Informatics, King’s College London, London WC2R 2LS, UK; 3NIHR Maudsley Biomedical Research Centre (BRC) at South London and Maudsley NHS Foundation Trust, King’s College London, London WC2R 2LS, UK; 4Genetics and Genomic Medicine, Great Ormond Street Institute of Child Health, University College London, London WC1E 6BT, UK; 5Biomedical Research Centre, NIHR Great Ormond Street Hospital, University College London, London WC1E 6BT, UK; 6Department of Clinical Genetics, Great Ormond Street Hospital, London WC1N 3JH, UK; 7Department of Clinical Genetics, Guy’s and St Thomas NHS Foundation Trust, London SE1 7EH, UK; 8Department of Medical Statistics, London School of Hygiene and Tropical Medicine, London WC1E 7HT, UK; 9King’s College Hospital, Bessemer Road, London SE5 9RS, UK

**Keywords:** genetic screening, genomics, next-generation sequencing, health informatics, mathematical model, Bayesian probability tools

## Abstract

**Importance:** Genomic sequencing enables the rapid identification of a breadth of genetic variants. For clinical purposes, sequencing for small genetic variations is considered a solved problem, while challenges remain for structural variants, given the lower sensitivity and specificity. Interest has recently risen among governing bodies in developing protocols for population-wide genetic screening. However, usefulness is constrained when the probability of being affected by a rare disease remains low, despite a positive genetic test. This is a common scenario in neurodegenerative disorders. The problem is recognised among statisticians and statistical geneticists but is less well-understood by clinicians and researchers who will act on these results, and by the general public who might access screening services directly without the appropriate support for interpretation. **Observations:** We explore the probability of subsequent disease following genetic screening of several variants, both single nucleotide variants (SNVs) and larger repeat expansions, for two neurological conditions, Huntington’s disease (HD) and amyotrophic lateral sclerosis (ALS), comparing these results with screening for phenylketonuria, which is well-established. The risk following a positive screening test was 0.5% for *C9orf72* in ALS and 0.4% for *HTT* in HD when testing repeat expansions, for which the test had sub-optimal performance (sensitivity = 99% and specificity = 90%), and 12.7% for phenylketonuria and 10.9% for ALS *SOD1* when testing pathogenic SNVs (sensitivity = 99.96% and specificity = 99.95%). Subsequent screening confirmation via PCR for *C9orf72* led to a 2% risk of developing ALS as a result of the reduced penetrance (44%). **Conclusions and Relevance:** We show that risk following a positive screening test result can be strikingly low for rare neurological diseases, even for fully penetrant variants such as *HTT*, if the test has sub-optimal performance. Accordingly, to maximise the utility of screening, it is vital to prioritise protocols with very high sensitivity and specificity, and a careful selection of markers for screening, giving regard to clinical interpretability, actionability, high penetrance, and secondary testing to confirm positive findings.

## 1. Introduction

Single nucleotide variants (SNVs) and small insertions and deletions (indels) are now genotyped in next-generation sequencing data with over 99% accuracy [1,2,3], while genotyping larger, structural variants is often less reliable [4]. For clinical purposes, the validity of sequencing for small genetic variations is considered a solved problem, while challenges remain for structural variants.

The widened availability of sequencing data has favoured our understanding of the role of genetic factors in various phenotypes [5,6,7,8]. Genetic testing has, therefore, become valuable for healthcare and has gained popularity among consumers who can access it directly, without advice from trained clinicians to guide the interpretation of results [9,10,11].

Interest has recently risen among governing bodies in developing protocols for population-wide genetic screening; such initiatives are being rolled out in the UK and are being considered in the US [12,13,14,15,16,17,18]. Genetic screening involves testing a population for genetic variants indicative of risk for specific diseases to identify people with either a higher predisposition of developing that disease or the potential to pass it on to their offspring. This approach utilises modern sequencing techniques to evaluate multiple genes associated with selected traits. In contrast, ‘targeted’ tests are those performed because of some suggestion that a person may harbour disease variants (e.g., symptoms or family history of disease). Although screening is relevant to liabilities ranging between monogenic and polygenic (cf. [16,19]), we focus here on screening for pathogenic variants with monogenic associations with rare diseases, particularly as applied to neurodegenerative disorders.

Although no widespread implementation of genetic screening protocols currently exist internationally, comparable metabolic screening testing neonates for metabolite markers of metabolic diseases is routine in many countries [17,20]. Positive metabolic tests are typically validated with secondary testing, including targeted genetic tests [9,21,22].

The utility of genetic screening can be assessed by the extent to which action can be taken following a positive test: its actionability [23,24]. One key tenet of actionability is the probability of having or later developing a disease following a positive test. Yet post-test disease probability can be strikingly low where disease risk prior to testing is low, as would be for population screening for rare diseases [25].

Bayesian inference, which is routine within clinical decision making [26], can be used to understand post-test ‘posterior’disease risk. Under this logic, disease probability following a test can be inferred given existing knowledge of the probability of other relevant events. Key considerations to understanding post-test disease risk, beyond pre-test (also known as ‘prior’) disease risk, include the genetic marker penetrance, its frequency among people displaying disease symptoms, and the sensitivity and specificity of the test (analytic validity). This reasoning is therefore highly relevant to screening for rare neurodegenerative diseases, for which genetic causes are typically rare variants of variable penetrance [27].

This article provides an overview of important considerations for genetic screening of rare disorders and it presents several case studies focused on neurodegenerative diseases. Considering conditional probability in medical decision making is not novel but these concepts must be emphasised in the genomic medicine era, since many results that could be obtained within large-scale indiscriminate testing of genetic variation across a population will not be actionable and may be misinterpreted. We modelled genetic screening for Huntington’s disease [28], HD, and amyotrophic lateral sclerosis [29], ALS, using Bayesian logic to examine the probability of disease following a positive test result for dichotomous ALS and HD genetic markers. We additionally modelled screening for phenylketonuria (PKU) [30] to compare genetic and metabolic screening.

## 2. Methods

### Definition of Probability in the Bayesian Framework

We used Bayesian logic to calculate the probability of having or subsequently manifesting disease *D* following a test result, indicating the presence or absence of marker *M* (the genetic variant). *M* is associated with increased liability of *D*, and positive test result *T* indicates the presence of *M*, while a negative test result *T*′ indicates its absence, denoted *M*′. Disease risk following a positive result is denoted by *P*(*D*|*T*), and *P*(*D*|*T*′) is used for a negative result.

Appendix A summarises the underlying logic. We assume that all model parameters represent binary events. This was a necessary simplification of reality as, for example, disease severity is not considered.

We estimated *P*(*D*|*T*) and *P*(*D*|*T*′) using the following input parameters:*P*(*D*), probability of a person having or later manifesting disease *D* prior to testing;*P*(*M*|*D*), frequency of marker *M* among those affected by *D*;*P*(*D*|*M*), penetrance, probability of having or later manifesting *D* for people harbouring *M*;*P*(*T*|*M*), sensitivity (true positive rate) of the testing procedure for detecting *M*;*P*(*T*′|*M*′), specificity (true negative rate) of the testing procedure for identifying the absence of *M*.

Bayes’ theorem was applied to derive the total probability of harbouring disease marker *M*.(1)$$P\left(M\right)=\frac{P(D)\times P\left(M|D\right)}{P\left(D|M\right)}$$and of disease *D* manifesting given the absence of *M*,(2)$$P\left(D|M^{\prime }\right)=\frac{P\left(D\right)\times \left(1-P\left(M|D\right)\right)}{\left(1-P\left(M\right)\right)}\;$$

We next calculated the total probability of positive test result *T*, *P*(*T*), according to the sensitivity and specificity of the test and the probabilities of *M* being present versus absent:(3)$$P\left(T\right)=P\left(T|M\right)\times P\left(M\right)+\left(1-P\left(T\prime |M\prime \right)\right)\times \left(1-P\left(M\right)\right)$$

Bayes’ theorem was then used to derive the probabilities of *M* being present after a positive result:(4)$$P\left(M|T\right)=\frac{P\left(M\right)\times P(T|M)}{P(T)}$$and for a negative result:(5)$$P\left(M|T^{\prime }\right)=\frac{P\left(M\right)\times \left(1-P\left(T|M\right)\right)}{\left(1-P\left(T\right)\right)}$$

The probabilities of manifesting disease *D* (which has conditional independence from *T* when considering *M*) after receiving a positive test result *T* were calculated as follows:(6)$$P\left(D|T\right)=P\left(D|M\right)\times P\left(M|T\right)+P\left(D|M^{\prime }\right)\times \left(1-P\left(M|T\right)\right)$$and for a negative test result *T*′, they were calculated with the following equation:(7)$$P\left(D|T\prime \right)=P\left(D|M\right)\times P\left(M|T^{\prime }\right)+P\left(D|M^{\prime }\right)\times \left(1-P\left(M|T^{\prime }\right)\right)$$

## 3. Case Studies

The Project MinE [31], ALS variant server [32], ClinVar [33], and gnomAD databases were searched alongside the previous databases and next-generation sequencing tool benchmarking reports to obtain data for the included case studies. Most input parameters were defined using data from the published literature and online databases. Sensitivity (*P(T|M)*) and specificity (*P(T|M)*) were defined by performance benchmarks for variant calling with state-of-the-art genomic sequencing techniques specialised for genotyping particular variant types (see Appendix A). For short tandem repeat expansion (STRE), we opted to base our modelling on the performance of short-read sequencing data because, although long-read sequencing could potentially offer higher performance in genotyping STREs, its higher cost and lower performance in genotyping SNVs make it a less practical choice for screening rare diseases compared to short-read sequencing [34,35]. Although analytical accuracy will vary across the genome and other sources of error exist, these heuristics were sufficient for our purposes.

Table 1 presents the parameter estimates and post-test disease risks calculated across various scenarios. A comprehensive description of parameter ascertainment, including penetrance estimation, is given in Appendix A summarises the assumptions and corresponding reality.

### 3.1. Case 1—Huntington’s Disease

HD is a late-onset Mendelian disease with autosomal dominant inheritance caused by a trinucleotide, CAG, STRE in the *HTT* gene (OMIM: 613004). We let *M* be a CAG expansion of >40 repeat units, which would have complete penetrance in a normal lifespan [28].

We modelled two scenarios for this example: (1) as in genetic screening, defining pre-test disease probability by using the baseline risk of HD in a general population and (2) as a targeted test, considering pre-test disease probability for an individual whose parent harbours the fully penetrant *HTT* STRE and who has a 0.5 probability of inheriting *M* (we did not model genetic anticipation [36]).

### 3.2. Case 2—Amyotrophic Lateral Sclerosis

ALS is a late-onset disease with locus and allelic heterogeneity; the genetic associations with risk and phenotype modification range between monogenic and polygenic. Variants in at least 40 genes are associated with ALS [29,37,38,39]. *SOD1* (OMIM: 147450) and *C9orf72* (OMIM: 614260) are the most frequently implicated genes, where variants of each account for fewer than 10% of cases. Autosomal dominant inheritance is typical for most people with a known genetic disease cause.

We modelled several definitions of markers for ALS risk, drawing from three of the commonest ALS genes: *SOD1*, *C9orf72*, and *FUS* (OMIM: 137070). *SOD1*- and *FUS*-linked ALS is typically attributed to SNVs and many pathogenic variants, with varying strength of supporting evidence reported in these genes [40]. The pathogenic form of *C9orf72* is a hexanucleotide, GGGGCC, STRE associated principally with the onset of either one or both ALS and frontotemporal dementia [41]. Known variants in these genes have typically incomplete penetrance; examples include ~90–100% penetrance for *SOD1* p.A5V and ~45% for the *C9orf72* STRE [42,43].

The definitions of *M* modelled in this case study were as follows:*SOD1* (all)—*M* includes any rare variant reported in people with ALS of European ancestry contained within the meta-analysis sample set from which the variant frequencies were derived (see Appendix A) [38];*SOD1* (A5V)—*M* represents the pathogenic *SOD1* p.A5V variant, one of the most common *SOD1* variants among North American ALS populations, characterised by high penetrance [42,44];*FUS* (all)—*M* includes any rare variant reported in people with ALS of European ancestry contained within the meta-analysis sample set from which the variant frequencies were derived (see Appendix A) [38];*FUS* (ClinVar)—*M* includes any of 21 *FUS* variants reported as pathogenic or likely pathogenic for ALS within ClinVar and present within databases of familial and sporadic ALS (see Appendix A) [31,32,33];*C9orf72*—*M* represents a pathogenic *C9orf72* STRE of 30≤ repeat GGGGCC units within the first intron of the *C9orf72* gene.

In the *SOD1* (all) and *FUS* (all) scenarios, *M* encompasses variants classed as pathogenic, likely pathogenic, and variants of uncertain significance [45]. It is appropriate to model these scenarios since many variants of uncertain significance in ALS-implicated genes have a high probability of being deleterious and should not necessarily be ignored [39].

For the *C9orf72* marker, we modelled two scenarios: (1) genetic screening with sensitivity and specificity defined by the performance of existing tools for genotyping STREs in sequencing data and (2) using repeat-primed polymerase chain reaction with amplicon-length analysis [46] as a secondary test to validate a positive result from screening via sequencing in scenario 1.

**Table 1 biomedicines-13-01018-t001:** Input parameters and disease risk estimates following testing for all case study scenarios. Parameter ascertainment is comprehensively described within Appendix A. HD = Huntington’s disease; ALS = amyotrophic lateral sclerosis; PKU = phenylketonuria. SNV = single nucleotide variant; STRE = short tandem repeat expansion. **^§^** Estimates are based on populations of predominantly European ancestry—95% confidence intervals shown for newly derived estimates in the ALS case study; * includes FUS variants classified as pathogenic or likely pathogenic for ALS in the ClinVar database and recorded within ALS population databases (see Appendix A); ^¶^ defined by variant-calling performance benchmarks of tools for genotyping in sequencing data by variant type (see Appendix A) and, where marked ^†^, by aggregate laboratory accuracy for genotyping *C9orf72* STRE with repeat-primed polymerase chain reaction and amplicon-length analysis [46]. ^Ω^ Risk following positive results in primary screening and confirmatory tests relative to a negative screening result (probability of PKU given a negative metabolic screening result is approximated as 1 × 10^−6^). Probabilities are defined in the methods.

Case Study	Gene Containing Marker (Case Study Scenario)	Variant Type	Pre-Test Disease Probability ^§^	Marker Frequency in People Affected ^§^	Penetrance ^§^	Test Sensitivity ^¶^	Test Specificity ^¶^	Disease Risk After Positive Test	Disease Risk After Negative Test	Relative Disease Risk After Positive Rather than Negative Test
-	-	-	P(D)	P(M|D)	P(D|M)	P(T|M)	P(T’|M’)	P(D|T)	P(D|T′)	-
**1:** **HD**	*HTT* (screening)	STRE	0.000410	1.000	1.000	0.990	0.900	0.00404	0.00000456	887
	*HTT* (targeted)	STRE	0.500	1.000	1.000	0.990	0.900	0.908	0.011	82.7
**2: ALS**	*SOD1* (all)	SNV	0.00333	0.0188 (0.0138, 0.0238)	0.701 (0.491, 0.926)	0.9996	0.9995	0.109	0.00327	33.3
	*SOD1* (A5V)	SNV	0.00333	0.000529 (4.43 × 10^−5^, 0.00101)	0.91	0.9996	0.9995	0.00683	0.00333	2.05
	*FUS* (all)	SNV	0.00333	0.00425 (0.0023, 0.0062)	0.579 (0.291, 0.884)	0.9996	0.9995	0.0302	0.00332	9.09
	*FUS* (ClinVar *)	SNV	0.00333	0.00251 (0.00125, 0.00377)	0.536 (0.211, 0.877)	0.9996	0.9995	0.0194	0.00333	5.84
	*C9orf72*	STRE	0.00333	0.0635 (0.0538, 0.0732)	0.439 (0.358, 0.520)	0.990	0.900	0.00519	0.00313	1.66
	*C9orf72* (positive sequencing screening confirmation)	STRE	0.0052	0.0635 (0.0538, 0.0732)	0.439 (0.358, 0.520)	0.95 ^†^	0.98 ^†^	0.0198	0.00489	4.06 (6.35 ^Ω^)
**3: PKU**	*PAH* (screening)	SNV	0.000100	0.743	0.892	0.9996	0.9995	0.127	0.0000257	4.961
	*PAH* (positive metabolic screening confirmation)	SNV	0.167	0.743	0.892	0.9996	0.9995	0.889	0.0497	17.9 (889,000 ^Ω^)

### 3.3. Case 3—Phenylketonuria

PKU is an autosomal recessive disease with infantile onset, caused by variants in the *PAH* gene (OMIM: 612349) [30]; most pathogenic variants for PKU are SNVs. *M* represents being homozygous or compound heterozygous for any of the three most common *PAH* variants recorded in European populations of people with PKU: p.Arg408Trp, c.1066-11G>A, p.Arg261Gln [30].

We modelled two testing scenarios for PKU: (1) genetic screening, with pre-test disease probability defined per the baseline population risk of PKU, and (2) secondary testing to confirm positive results obtained using tandem mass spectrometry [47] as in established metabolic screening protocols (see Appendix A).

## 4. Results and Discussion

### 4.1. Post-Test Disease Probability

#### 4.1.1. Screening Versus Diagnostic Testing

Across the case studies, we showed a low probability of disease following positive results within contextually blind genetic screening scenarios; risk ranged between 12.7% and 0.4% (see Table 1). Disease risk was always negligible following a negative test result, indicating absence of the tested variant, as would be seen for any rare trait.

The HD case study illustrates the distinction between contextually blind screening and diagnostic testing for rare diseases: following a positive test result, lifetime HD risk was high (90.8%) using targeted testing but low (0.4%) with screening. This difference reflects the effect of the sub-optimal performance (sensitivity = 99% and specificity = 90%) of the test for repeat expansions generating a large number of false positives in population screening despite the complete penetrance of *HTT*. Indeed, unlike screening, targeted testing is performed based on some indication of a person’s elevated disease risk (e.g., existing disease symptoms or family history). Accordingly, pre-test disease probability is greater. Inherently low pre-test disease probability will be a pervasive issue in screening for rare diseases.

#### 4.1.2. Relative Risk and Secondary Testing

The utility of a test for identifying at-risk individuals can be examined based on relative disease risk following positive rather than negative test results: utility is limited when risk is only minimally greater for people testing positive rather than negative. This is observed in the ALS *C9orf72* case study, where risk was only 1.7 times greater (Table 1) following a positive screening from sequencing alone, despite this variant being the most common genetic cause of ALS [38].

We additionally observed 6.35 times greater ALS risk for a person testing positive on both screening for *C9orf72* and a secondary test than for a person testing negative on the initial screening. This increased relative risk reflects that a person testing positive on two independent measures of disease risk has a greater absolute probability of disease than after the initial screening result alone.

We emphasise that secondary testing is important to increase certainty in positive tests. The PKU case study demonstrates its potentially sizable impact. A positive screening result using the established metabolic approach alone indicated 16.7% PKU risk, versus 12.7% with genetic screening. The metabolic marker, which is universal across people with PKU and indicates existing disease manifestation, eclipses the need for genetic screening for PKU, marked by variants with incomplete penetrance that are not present for all people with PKU. However, the genetic test remains useful for validating the positive metabolic screening result [17]: the probability of PKU following a confirmatory genetic test conducted on the basis of a positive metabolic screening result was 88.9%.

The overall benefit of secondary testing will, however, differ by scenario. For ALS, the risk remained moderate (~2%) despite two positive test results for *C9orf72*.

#### 4.1.3. Constraints upon Post-Test Disease Probability

Figure 1, Figure 2 and Figure 3 illustrate how the post-test disease probability reduces as the probability of any test, disease, or marker characteristic decreases. Sensitivity and specificity critically constrain certainty about post-test disease risk, and this role is amplified as the other parameter probabilities decrease. Figure 1 particularly demonstrates the increased role of specificity in rarer diseases, where disease risk following a positive test result will be moderated only by penetrance in a protocol of perfect specificity.

M occurs in all people with D (P(M|D) = 1) and has complete penetrance (P(D|M) = 1)). The plot lines are defined according to the sensitivity (P(T|M)) and specificity (P(T′|M′)) of existing protocols for genotyping variant types (see Appendix A): single nucleotide variant (SNV), test sensitivity P(T|M) = 0.9996, and test specificity P(T′|M′) = 0.9995; small insertion or deletion (Indel) and test sensitivity P(T|M) = 0.9962, test specificity P(T′|M′) = 0.9971; short tandem repeat expansion (STRE), test sensitivity P(T|M) = 0.99, and test specificity P(T′|M′) = 0.90; copy number variant (CNV)-del. (deletion), test sensitivity P(T|M) = 0.289, and test specificity P(T′|M′) = 0.959; and CNV-dup. (duplication), test sensitivity P(T|M) = 0.1020, and test specificity P(T′|M′) = 0.9233. The probabilities are defined in the methods.

Figure 2 illustrates the impact of the genetic test’s performance (sensitivity and specificity) in existing protocols for genotyping different variant types (SNVs, indels, STREs, and CNVs) on the probability of disease, given a positive test result. As is well-recognized, high sensitivity and specificity are crucial for maximizing the utility of testing, especially for rare diseases. The respective trade-offs between prioritising each of these must be regarded: high sensitivity is required to detect markers, while high specificity increases confidence in positive results. Established screening protocols prioritise high sensitivity to maximise detection of at-risk individuals, with confirmatory secondary testing being vital to minimise false-positive results [9,21,22].

Since the characteristics of diseases and associated variants are all pre-determined within a population, the disease markers (variants) screened should be chosen carefully. Figure 3 illustrates how the prevalence of a marker among affected individuals, its penetrance, and the performance of the test (sensitivity and specificity) influence the probability of developing the disease given a positive test result. The most useful variants are those that are more prevalent among affected individuals, have high penetrance, and can be genotyped with high sensitivity and specificity (see examples of real and hypothetical diseases and markers in Figure 3A–D).

### 4.2. Practical Implementation of Genetic Screening

#### 4.2.1. Marker Selection

Before a marker is used in screening, its relevance across people must be evaluated, recognising that this may vary by ancestry. For instance, particular variants may be less common or only present in certain populations, and penetrance can also vary between them [37,44]. Screening protocols must therefore account for these differences to prevent systemic inequalities, especially for minorities, which are often under-studied and which therefore have limited genetic information available.

Regard must be given to the clinical interpretability of selected markers. We illustrated several approaches to defining markers in the ALS case study. Within the *SOD1* (all) scenario, disease risk is marked by an aggregation of putatively pathogenic *SOD1* variants. Without curation, the relationship to disease varies across them, e.g., *SOD1* p.I114T has substantially lower penetrance than p.A5V, and many potentially relevant variants have uncertain significance [33,39,40,43]. Curation could involve defining a positive result as presence of variants with sufficient evidence of pathogenicity [45], as in the *FUS* (ClinVar) scenario, or as harbouring specific variants, as in the *SOD1* (A5V) scenario.

De novo variants and variants of uncertain significance present a substantial challenge for screening since they will frequently be identified, yet must be set aside until variant interpretation is possible despite potentially being deleterious [12]. PKU demonstrates the scale of this issue for rare diseases with multiple implicated variants, as 55% of deleterious *PAH* genotypes are observed uniquely [30].

#### 4.2.2. Utility over Time and Actionability

As genetic screening is possible from birth, while non-genetic methods may not be, the age of viability for screening methods should be evaluated. For late-onset diseases, early genetic screening may enable preventative treatments to at-risk individuals or close monitoring for prodromal disease markers. For instance, rapid eye movement sleep behaviour disorder is a prodromal feature for Parkinson’s disease [48], and the monitoring of at-risk individuals may enable early intervention. The influence of time upon treatment viability and effectiveness must also be considered. For example, genetic therapy has potential utility for preventing or delaying the onset of degenerative disorders [49].

The ultimate benefit of early identification of disease risk through genetic screening is contingent upon the actionability of the result. A framework of actionability [23,24], shown to align with laypersons’ views on treatment acceptability [50], states that actionability is determined by disease likelihood and severity, intervention effectiveness in disease minimisation or prevention, and the consequence of the intervention to a person and risk if not performed. Each of these elements are critical when selecting traits and markers for genetic screening and for the clinical interpretation of results.

### 4.3. Limitations

The Bayesian logic in the case studies simplifies genotype–phenotype relationships and cannot address all considerations relevant to clinical genetic testing. Phenotype variability is not considered. Other factors include polygenicity and oligogenicity, pleiotropy, the role of genetic and environmental modifying factors, and additive genetic effects in recessive conditions and heterozygous carriers of pathogenic variants. Such influences can fundamentally impact both the probability that a disease will manifest and its severity. For instance, although we used widely accepted thresholds to identify pathogenic repeat expansions in this study (≥30 repeats for *C9orf72* and >40 repeats for *HTT*), it is important to note that the CAG repeat length in *HTT* shows a strong negative correlation with age at symptom onset [51]. Additionally, *C9orf72* repeats could contribute to oligogenic forms of ALS, and intermediate lengths may have a smaller effect on disease risk [52,53]. Any results from genetic screening must be interpreted within the wider context of that disease and its modifiers.

## 5. Conclusions

We have shown that risk following a positive screening test result can be strikingly low for rare neurological diseases. Accordingly, to maximise the utility of screening, we suggest prioritising protocols of very high sensitivity and specificity, careful selection of markers for screening, giving regard to clinical interpretability and secondary testing to confirm positive findings.

A key advantage of a genetic screening approach for late-onset diseases is that these markers can be examined across the lifespan. Hence, positive test results could be useful for targeting people for prevention and for the monitoring of prodromal features.

Although considering disease risk within a Bayesian context is not novel, it is important to stress that the considerations raised here come at a time when governments are evaluating the implementation of genomic sequencing for population screening and as access to genetic testing outside healthcare settings increases. While genetic screening has many potential benefits, the limitations of such an approach should be properly understood. Policy makers must consider the impact of a positive test result on large numbers of people that will never develop a given disease, a particularly salient issue for late-onset diseases. Although not the present focus, the substantial ethical and social considerations raised in conjunction with screening must also be regarded [13].

## Figures and Tables

**Figure 1 biomedicines-13-01018-f001:**
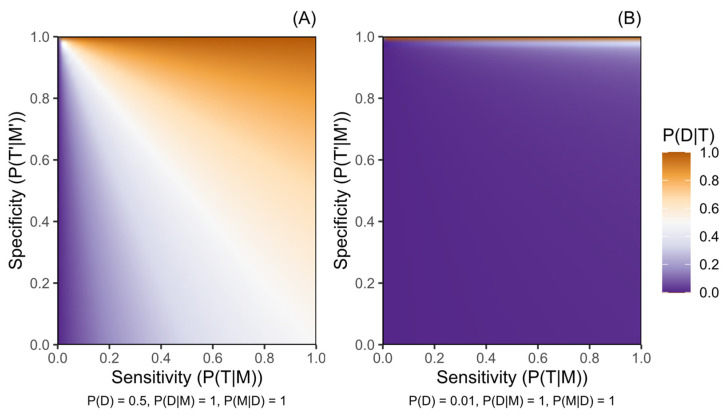
Probability of a disease given a positive genetic test result for a marker of increased disease risk (P(D|T)) according to the sensitivity (P(T|M)) and the specificity (P(T′|M′)) of the testing protocol. (**A**) presents this for a disease with pre-test probability (P(D)) of 0.5, while (**B**) presents a disease with P(D) of 0.01. Penetrance is complete (P(D|M) = 1) and variant M is harboured by all people with disease D (P(M|D) = 1) in both panels. Probabilities are defined in the methods.

**Figure 2 biomedicines-13-01018-f002:**
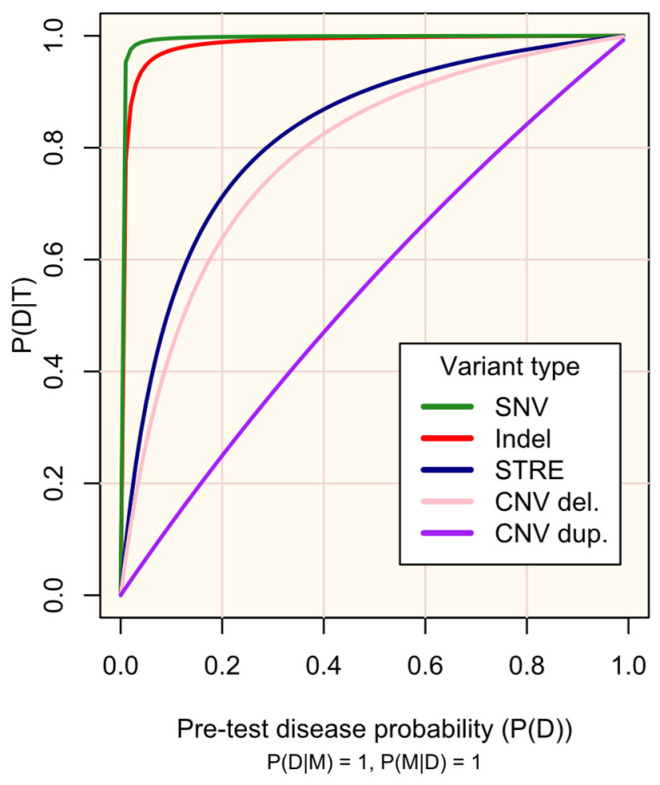
Probability of disease D following a positive genetic test result for marker M (P(D|T)) according to pre-test disease probability (P(D)).

**Figure 3 biomedicines-13-01018-f003:**
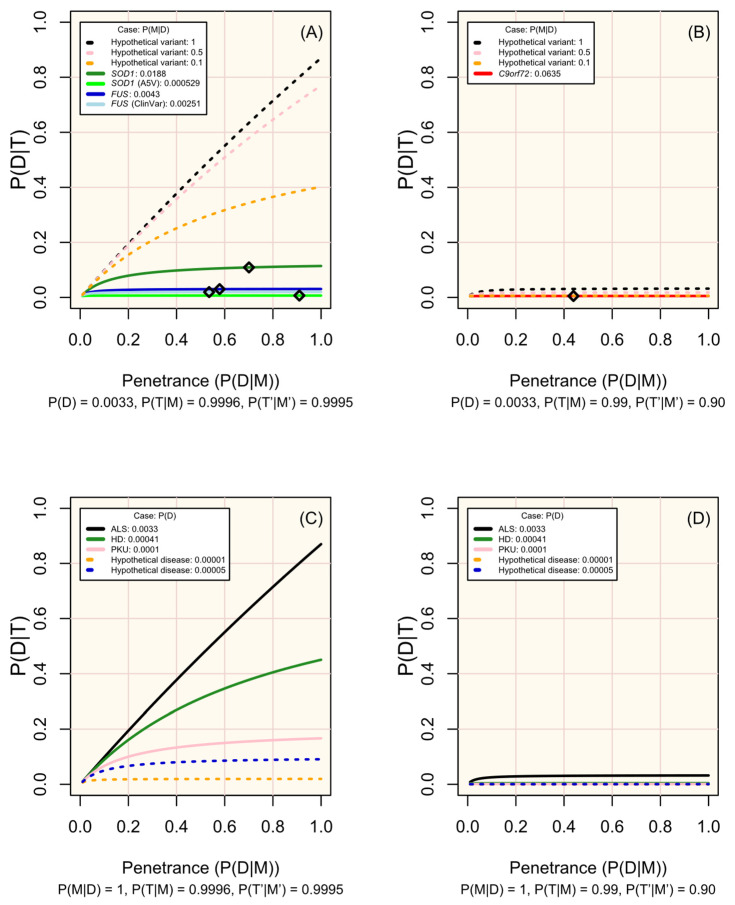
Change in disease risk following a positive test result for a marker of increased disease risk (P(D|T)) according to penetrance (P(D|M)). (**A**,**B**) display modelled and hypothetical markers of ALS which differ in frequency across people affected by ALS (P(M|D)), where pre-test disease probability (P(D)) is 0.0033, and diamonds mark the penetrance estimated for non-hypothetical variants (see Table 1). (**C**,**D**) display diseases in which P(M|D) = 1 and with P(D) set in line with the modelled case studies or a hypothetical rare disease. Analytic validity parameters are defined according to the performance of sequencing tools for genotyping single nucleotide variants in (**A**,**C**), and of short tandem repeat expansions in (**B**,**D**) (see Appendix A; Figure 2). Probabilities are defined in the methods.

## Data Availability

Data are contained within the article.

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
