# Peer review of "Modelling Population Genetic Screening in Rare Neurodegenerative Diseases"

_biomedicines, 2025, doi:10.3390/biomedicines13051018_

Round 1
Reviewer 1 Report
Comments and Suggestions for Authors
I have reviewed the manuscript Spargo et al., title "Modelling population genetic screening in rare neurodegenerative diseases" on genetic screening for neurological conditions. The paper addresses an important topic, the challenges of interpreting the positive genetic test results in rare diseases but it requires significant revision before publication.
The fundamental premise of the manuscript is valuable, for example, positive predictive values can be low for rare conditions even with good tests. The analysis of C9orf72 in ALS (0.5% risk, increasing to 2% after PCR confirmation) and SOD1 variants (10.9% risk) are reasonable given the known incomplete penetrance of these genetic factors. Similarly, the phenylketonuria analysis (12.7% risk) aligns with established screening principles and serves as an appropriate comparison.
However, the specific analysis of Huntington's disease (HD) screening contains flaws that undermine part of the conclusions in the current article. HD is characterized by near-complete penetrance for full mutations >40 CAG repeat expansion, age-dependent onset related to CAG repeat length, and 50% inheritance probability in affected families. The reported 0.4% risk of HD following a positive test contradicts established literature showing that individuals with confirmed expansions above 40 repeats develop the disease with hundred percent certainty if they live long enough.
I recommend either, removing the HD analysis entirely, as the paper's central thesis is well-supported by the ALS and phenylketonuria examples alone, or major revision of the HD section with input from specialists in HD genetics
Author Response
Reviewer 1:
I have reviewed the manuscript Spargo et al., title "Modelling population genetic screening in rare neurodegenerative diseases" on genetic screening for neurological conditions. The paper addresses an important topic, the challenges of interpreting the positive genetic test results in rare diseases but it requires significant revision before publication.
The fundamental premise of the manuscript is valuable, for example, positive predictive values can be low for rare conditions even with good tests. The analysis of C9orf72 in ALS (0.5% risk, increasing to 2% after PCR confirmation) and SOD1 variants (10.9% risk) are reasonable given the known incomplete penetrance of these genetic factors. Similarly, the phenylketonuria analysis (12.7% risk) aligns with established screening principles and serves as an appropriate comparison.
However, the specific analysis of Huntington's disease (HD) screening contains flaws that undermine part of the conclusions in the current article. HD is characterized by near-complete penetrance for full mutations >40 CAG repeat expansion, age-dependent onset related to CAG repeat length, and 50% inheritance probability in affected families. The reported 0.4% risk of HD following a positive test contradicts established literature showing that individuals with confirmed expansions above 40 repeats develop the disease with hundred percent certainty if they live long enough.
I recommend either, removing the HD analysis entirely, as the paper's central thesis is well-supported by the ALS and phenylketonuria examples alone, or major revision of the HD section with input from specialists in HD genetics
We thank the reviewer for the appreciation of our work and the comment regarding HTT in HD. We completely agree with the reviewer about the established full penetrance of HTT and apologise for the lack of clarity. Our result, 0.4% risk of HD given a positive test of HTT, is the consequence of the test’s sub-optimal performance for repeat expansion (sensitivity=99% and specificity =90%) and the rarity or HD which lead to a large number of false positives. We have stressed the full penetrance of HTT and that our result is the consequence of sub-optimal test performance in several parts of the text and we hope this is now acceptable for the reviewer. Some examples below:
1)In the abstract: The risk following a positive screening test was 0.5% for C9orf72 in ALS and 0.4% for HTT in HD, when testing repeat expansions for which the test had sub-optimal performance (sensitivity=99% and specificity =90%),…. We show that risk following a positive screening test result can be strikingly low for rare neurological diseases even for fully penetrant variants such as HTT if the test has sub-optimal performance….
In the methods:
2)…a CAG expansion of >40 repeat units, which would have complete penetrance in a normal lifespan (29)…
3) .. for an individual whose parent harbours the fully penetrant HTT STRE..;
4) Results and discussion: … following a positive test result, lifetime HD risk was high (90.8%) using targeted testing but low (0.4%) in screening. This difference reflects the effect of the sub-optimal performance (sensitivity=99% and specificity =90%) of the test for repeat expansions generating a large number of false positives in population screening despite the complete penetrance of HTT….

Reviewer 2 Report
Comments and Suggestions for Authors The authors tried to model the risk of certain neurological diseases following a positive test screen and suggested that population screening is unnecessary. Major concerns: Table 1 is between the texts without any labeling (if that is indeed Table 1). Figures are not explained at all in the manuscript, but only in the figure legends. If the authors could also explain their figures in the text, it would be easier for other people to understand. Probability and folds in the text are also not explained well. If the authors could mention either the equation or refer to the figures, those numbers would be more convincing. The authors should also check their references carefully. "Error! Reference source not found." is seen through the manuscript, which affects reading and understanding, and needs to be fixed. The case studies are a little confusing, especially for the ALS case, as there is a PAH table in the text. Are there more data similar to the phenylketonuria data for both ALS and HD? Minor issues: Check sentences to make sure they are all complete. For example, introduction p2 line3 "who can it directly"Author Response
Reviewer 2:
The authors tried to model the risk of certain neurological diseases following a positive test screen and suggested that population screening is unnecessary. Major concerns: Table 1 is between the texts without any labeling (if that is indeed Table 1). Figures are not explained at all in the manuscript, but only in the figure legends. If the authors could also explain their figures in the text, it would be easier for other people to understand. Probability and folds in the text are also not explained well. If the authors could mention either the equation or refer to the figures, those numbers would be more convincing. The authors should also check their references carefully. "Error! Reference source not found." is seen through the manuscript, which affects reading and understanding, and needs to be fixed. The case studies are a little confusing, especially for the ALS case, as there is a PAH table in the text. Are there more data similar to the phenylketonuria data for both ALS and HD? Minor issues: Check sentences to make sure they are all complete. For example, introduction p2 line3 "who can it directly"
We would like to thank the reviewer for highlighting these issues. We believe most of them are due to the submission system which messed up the article references, tables and figures. We have tried to fix these and we have uploaded a pdf version which seems to visualise well. Please see in the following how we have improved all things suggested by the reviewer:
1) Table 1 is between the texts without any labeling (if that is indeed Table 1):
This was indeed messed up by the system. Table one has an extensive legend and has labels. Please also find it at this link in case it is not visualised well on the journal website: https://github.com/AlfredoKCL/screening_paper_review/blob/main/Table1.pdf
2) Figures are not explained at all in the manuscript, but only in the figure legends.
We have added an explanation of the figures in the text:
Figure 1, Figure 2, and Figure 3 illustrate how the post-test disease probability reduces as the probability of any test, disease, or marker characteristic decreases. Sensitivity and specificity critically constrain certainty about post-test disease risk, and this role is amplified as the other parameter probabilities decrease.
Figure 1 particularly demonstrates the increased role of specificity in rarer diseases, where disease risk following a positive test result will be moderated only by penetrance in a protocol of perfect specificity.
Figure 2 illustrates the impact of the genetic test's performance (sensitivity and specificity) in existing protocols for genotyping different variant types (SNVs, indels, STREs and CNVs) on the probability of disease, given a positive test result. As is well recognized, high sensitivity and specificity are crucial for maximizing the utility of testing, especially for rare diseases.
Figure 3 illustrates how the prevalence of a marker among affected individuals, its penetrance, and the performance of the test (sensitivity and specificity) influence the probability of developing the disease given a positive test result. The most useful variants are those that are more prevalent among affected individuals, have high penetrance, and can be genotyped with high sensitivity and specificity (see examples of real and hypothetical diseases and markers in Figure 3, panels A-D).
3) Probability and folds in the text are also not explained well. If the authors could mention either the equation or refer to the figures, those numbers would be more convincing.
Probabilities in our Bayesian framework are defined in the methods. To make this clearer we have renamed the section of the methods where these are defined as “Definition of probability in the Bayesian framework”. We have also added a reminder about where these are defined in all figure and table legends that report and display probabilities.
4) The authors should also check their references carefully. "Error! Reference source not found." is seen through the manuscript, which affects reading and understanding, and needs to be fixed.
This was fixed. However, we believe the system has messed up our references. Please find an intact version of the paper in pdf format here https://github.com/AlfredoKCL/screening_paper_review/blob/main/article.pdf
5) The case studies are a little confusing, especially for the ALS case, as there is a PAH table in the text. Are there more data similar to the phenylketonuria data for both ALS and HD?
Apologies, this was again due to the system miss-formatting table 1. It should be fixed but you can also find Table 1 here https://github.com/AlfredoKCL/screening_paper_review/blob/main/Table1.pdf
6) Minor issues: Check sentences to make sure they are all complete. For example, introduction p2 line3 "who can it directly":
Fixed.

Round 2
Reviewer 1 Report
Comments and Suggestions for Authors
Thank you for your response to my previous comments, I appreciate the clarifications you have made regarding HD and the penetrance of HTT expansions. You have successfully addressed my concerns by explicitly acknowledging throughout the manuscript that HTT expansions >40 CAG repeats have complete penetrance. The revisions you’ve implemented make it clear that the low positive predictive value (0.4%) calculated for population screening is a consequence of HD’s rarity combined with the assumed test performance characteristics (99% sensitivity, 90% specificity), rather than an indication of reduced penetrance. Your revised manuscript provides valuable insights into the challenges of interpreting positive genetic test results in rare neurological conditions. The multiple examples you present (C9orf72 and SOD1 in ALS, HTT in HD, and the comparison with phenylketonuria) collectively build a compelling case for careful consideration of test characteristics and disease prevalence when implementing genetic screening programs.
Author Response
We would like to thank the reviewer for the appreciation of our work, for the effort they put in reviewing it and for proposing valid and useful comments that allowed us to improve our manuscript.
Reviewer 2 Report
Comments and Suggestions for Authors
The updated manuscript demonstrates the results better than the earlier version and is easier to follow.
One comment about the result is that the threshold the authors used, which is widely accepted, might cause some issues for disease modeling. For example, if C9 expansion carriers have thousands of repeats may have a different probability of developing the disease within their lifespan than other carriers with only 31 repeats. This situation could also be true for HTT, which is also a repeat expansion disease. However, this would be the risk for the follow-up test. Maybe the authors could consider this factor in their model to see if there are any differences for people with different repeat lengths.
For screening, short-read or error-prone long-read sequencing could give false positive results, especially for people with near-threshold repeats. But accurate long-read sequencing would cost more money and wouldn't be worth it, given the low rate of disease probability. This might also affect the risk for HTT after a positive screening, which also suggests that population-based screening is not recommended. For SNV diseases, short-read sequencing is not expensive and accurate enough. So maybe the fundamental differences lie in the sequencing techniques used for these two different types of diseases.
Minor issue: The second figure is still Figure 1.
Author Response
We would like to thank the reviewer for their comments. The reviewer is correct about the lack of modelling of phenotype-genotype interactions. Unfortunately the Bayesian logic in our study simplifies genotype-phenotype relationships and cannot address all considerations relevant to clinical genetic testing such as phenotype variability. We have added some text in the Limitations section to discuss this:
"The Bayesian logic in the case studies simplifies genotype-phenotype relationships and cannot address all considerations relevant to clinical genetic testing. Phenotype variability is not considered. Other factors include: polygenicity and oligogenicity, pleiotropy, the role of genetic and environmental modifying factors, and that of additive genetic effects in recessive conditions and heterozygous carriers of pathogenic variants. Such influences can fundamentally impact both the probability that a disease will manifest and its severity. For instance, Although we used widely accepted thresholds to identify pathogenic repeat expansions in this study (≥30 repeats for C9orf72 and >40 repeats for HTT), it is important to note that the CAG repeat length in HTT shows a strong negative correlation with age at symptom onset (49). Additionally, C9orf72 repeats could contribute to oligogenic forms of ALS and intermediate lengths may have a smaller effect on disease risk (50, 51). Any results from genetic screening must be interpreted within the wider context of that disease and its modifiers."
Regarding the use of long-reads for genotyping STREs, we agree with the reviewer. The choice of technology has a huge impact on the results and unfortunately long-reads are not practical for screening of rare diseases. We have added the following text to the manuscript to allow the reader to have a better understanding of our choices their impact:
"For short tandem repeat expansion (STRE), we opted to base our modelling on the performance of short-read sequencing data because, although long-read sequencing could potentially offer higher performance in genotyping STREs, its higher cost and lower performance in genotyping SNVs make it a less practical choice for screening rare diseases compared to short-read sequencing (34, 35). "